# Experiences and Attitudes of Parents Reducing Carbohydrate Intake in the Management of Their Child’s Type 1 Diabetes: A Qualitative Study

**DOI:** 10.3390/nu15071666

**Published:** 2023-03-29

**Authors:** Amelia J. Harray, Alison G. Roberts, Naomi E. Crosby, Charlene Shoneye, Keely Bebbington

**Affiliations:** 1Children’s Diabetes Centre, Telethon Kids Institute, Perth, WA 6009, Australia; 2Medical School, The University of Western Australia, Perth, WA 6009, Australia; 3School of Population Health, Curtin University, Perth, WA 6102, Australia; 4Department of Endocrinology and Diabetes, Perth Children’s Hospital, Perth, WA 6009, Australia; 5Centre for Child Health Research, University of Western Australia, Perth, WA 6009, Australia

**Keywords:** type 1 diabetes, low carbohydrate diet, child, grounded theory, constant comparative analysis

## Abstract

Reducing carbohydrate (CHO) intake is being used as an approach to manage type 1 diabetes (T1D) in children. This study aimed to investigate the experiences and attitudes of parents of children with T1D who are reducing CHO intake to help manage blood glucose levels (BGLs). Semi-structured interviews were conducted with the parents of children with T1D for >1 year who reported implementing a low CHO approach to manage BGLs. Data were analysed using a constant comparative analysis approach. Participants (*n* = 14) were parents of children (6.6 ± 2.0 years) with T1D in Western Australia. All parents reported different methods of CHO restriction and all perceived that benefits outweighed challenges. Parents reported feeling less worried, had improved sleep and felt their child was safer when using a low CHO approach due to more stable BGLs. Reported challenges included: increased cost and time spent preparing food; perceived judgement from others; and child dissatisfaction with restricted food choices. Parents reported accessing information and support through social media networks. Parents reported a desire for more research into this approach. Understanding the promoters and barriers for this dietary approach may inform strategies to better engage and support families with approaches that align with current evidence while considering their concerns around safety and hyperglycaemia.

## 1. Introduction

Prior to the introduction of animal-derived insulin, restricting carbohydrate (CHO) intake was used to manage type 1 diabetes (T1D) [1]. Today, children with T1D are encouraged to eat a variety of nutritious foods with carbohydrates providing about 50% of one’s total daily energy intake [2]. However, low CHO diets are regaining worldwide popularity, with a common definition being less than 26% of one’s total daily energy intake from CHO [3,4]. In adults with T1D, low CHO diets have been shown to decrease postprandial hyperglycaemia, improve glycated haemoglobin (HbA1c), increase time in range and reduce hypoglycaemia risk due to reduced insulin requirements [5,6]. However, there are limited studies on the impact of low CHO diets on children, and international evidence-based nutrition guidelines for children with T1D recommend 45–55% of total daily energy intake from CHO, composed of grains and cereals, fruits and starchy vegetables [2,7]. Despite low CHO diets not being recommended, many families of children in Western Australia (WA) are reporting using this approach to manage their child’s T1D. A survey of 624 children with T1D in the Czech Republic found 39% had experimented with a low CHO diet and 6% were following a low CHO diet [8]. The authors found over half of those implementing a low CHO diet had not discussed the diet with their healthcare providers, highlighting the need for more research in this area.

Low CHO diets often involve replacing foods containing CHO (e.g., grains, cereals, fruit, milk) with foods that are high in protein and/or fat [9,10]. Where CHO are replaced with saturated fat and animal products, there are concerns about the increased risk of cardiovascular disease. Such dietary patterns raise concerns relating to childhood growth, bone health, nutritional status and increased risk of cardiovascular disease [9,10,11]. In addition, evidence suggests restrictive diets can increase the risk of disordered eating in childhood, adolescence and adulthood [12]. Given that type 1 diabetes itself is a risk factor for cardiovascular disease and eating disorders, research into this dietary pattern is needed. Other risks of nutritional and metabolic safety include mineral deficiencies and such as severe hypoglycaemia [13]. As popularity of low CHO diets for children with T1D increases, qualitative investigations to explore application of this dietary approach to manage diabetes are needed. This study aimed to investigate the experiences and attitudes of parents reducing their child’s CHO intake to help manage blood glucose levels.

## 2. Methods

This qualitative study applied a constructivist grounded theory approach using an interpretive paradigm. This was used to construct meaning from participants’ stories and develop a theory around decisions and influences around using lower CHO diets [14].

### 2.1. Participants

Parents and caregivers of young children with T1D who access Perth Children’s Hospital’s Diabetes Clinic are added to the population-based, Western Australian Children’s Diabetes Database (WACDD) for all children living with T1D in the state. Parents of children with T1D aged ≤10 years were sent invitations via email to participate in this study. The inclusion criteria included parents/caregivers implementing what they perceived to be a low CHO diet for their child with T1D (at time of interview or within the preceding 6 months). Potential participants were excluded if they had a lack of English proficiency to understand the interview questions and conversations, ascertained by the researcher when explaining the study via telephone.

Recruitment methods for this study included sending 262 initial emails to parents of children living with T1D. Social media channels and electronic newsletters, via the Children’s Diabetes Centre website, were also used to promote diverse recruitment. Participants were given time to consider the study requirements prior to providing written consent and were informed they could withdraw at any time.

### 2.2. Data Collection

Once consent was provided, parents were sent a link to an online REDCap survey to provide demographic and psychosocial information [15]. Questionnaires collected information on parents’ education level, employment and relationships, including marital status, relationship to child, siblings and number of people in the house. Questions were also asked about their child with T1D, including date of birth, diagnosis, family history of T1D, insulin regimen (multiple daily injections or continuous subcutaneous insulin infusion (CSII)), continuous glucose monitor (CGM) use and length of treatment modality. The child’s HbA1c within the preceding 3 months of the interview was obtained from the WACCD.

Semi-structured interviews were used to explore parents’ understanding and use of a low CHO approach to manage their child’s T1D. An interview guide (Appendix A) was developed based on the current literature and input from the research team in consultation with a community reference group of families with children living with T1D.

Parents were given the choice to complete the interview face-to-face, via telephone or via video call. However, early in recruitment, face-to-face interviews were no longer possible due to COVID-19 restrictions. Evidence suggests that face-to-face interviews may last longer and have more conversation turns, yielding more detailed conversations [16]. However, telephone interviews also provide rich data, enabling parents from remote areas, those isolating due to COVID and those that are too time poor to participate. To ensure consistency and encourage open conversation in all interviews, the same researchers (A.J.H., A.G.R.) conducted the interviews, using the interview guide. They were experienced health professionals trained in qualitative research and chronic disease but not part of the clinical team. Interviews were audio recorded with consent, transcribed, deidentified and checked for accuracy by three members of the research team.

### 2.3. Data Analysis

Data collection and initial analysis took place concurrently to ensure categories identified in early interviews could be explored further in subsequent interviews. All transcribed interviews were imported to NVivo (QRS International Pty Ltd., Boston, MA, USA, March 2020) to assist with data management, enabling the identification of codes and retrieval of participant quotes [17].

Data from interviews were analysed using a constant comparative analysis approach. This involved comparing similarities and differences and integrating excerpts of raw data into codes. These codes were used to create categories, which were continually refined until no new data were obtained [14]. This iterative process involved an inductive approach and was conducted by two independent researchers. The research team (A.J.H., A.G.R., N.E.C., C.S., K.B.) worked together to develop and discuss the final categories and generate a theory that were representative of the data. An audit trail was maintained to explain the process of analysis, including records of decision-making processes.

## 3. Results

Fourteen parents of children (*n* = 13) with T1D (mean age 6.6 ± 2.0 years) were recruited between December 2020 and March 2021. Zoom was used to complete individual interviews (*n* = 3) and one group interview (*n* = 2). Remaining individual interviews were completed face-to-face (*n* = 2) or on the telephone (*n* = 7) between December 2020 and March 2021. The interviews ranged from 37 min to 76 min (mean 54 ± 12 min). Continuous glucose monitors were used by 93% of families, enabling parents to track their child’s BGLs in real-time. Additional characteristics of parents and their children can be seen in Table 1.

Four integrated concepts were developed from the participants’ descriptions of their experiences: (1) understanding and implementation of low CHO diets; (2) promoters to implementing a low CHO diet; (3) barriers to implementing a low CHO diet; and (4) sources of information on low CHO diets. A thematic map of the key themes can be seen in Figure 1.

### 3.1. Understanding and Implementation of Low CHO Diets

Parents’ understanding of and strategies for implementing a low CHO diet varied, including reports of lowering of their child’s overall daily CHO intake, specifying CHO limits per day (ranging from 10–150 g/day) or per meal (e.g., 15–30 g/meal) or avoiding specific foods. The most commonly avoided source of CHO was from foods in the grains and cereals food group, as per the Australian Dietary Guidelines [9].


*“I avoid flour at all costs. If I do need to add flour, I’ll halve the amount I need to put in, and I’ll substitute the rest with almond meal, or coconut flour. I do a lot of that kind of stuff as well, err, just because flour’s just a killer, I feel.”*
(Participant 11)


*“So initially we were probably around 30 g per day of carbs, or little more with some snacks. Now, some days might be a little under some might be a bit over but usually I would say it’s still probably less than 50 g a day of carbohydrate, and that usually comes in the form of vegetables, some small amount in some cheeses, nuts.”*
(Participant 5)

Some parents reported a flexible low CHO approach, where only small changes were made, or children would eat lower CHO substitutes according to their preferences and appetite.


*“…I’ll let him eat the cakes and lollies and stuff that are there [party]. It’ll be one of those special treat days that he gets.”*
(Participant 11)


*“We do lot of alternative things… he does sometimes have bread but it’s a low carb bread, but then we do a lot more of the… wraps, things like that. We don’t bail everything out.”*
(Participant 12)

Implementing a low CHO diet at social occasions and parties was commonly reported as challenging. Some parents brought their own food or prepared low CHO variations (e.g., marshmallows, no-sugar ice cream) and others replaced lollies with toys or saved party bags to for future hypo treatments.


*“…little gift bag with some toys instead of lollies and she’s really happy with that.*
(Participant 10)

In contrast, other parents expressed the importance of keeping BGLs within a tight range regardless of social events so that incidental eating was restricted. 


*“yeah, or they’re like, even, like one, one grape its ok. Like the other day he was 5.6 and I was trying to get it back down to a bit more normal… “can’t he just have one piece of fruit?” but that fruit will take him from 5.6 to 7 or 8 and that’s when he’ll already feel bad.”*
(Participant 1)

The initial implementation of a low CHO diet was commonly reported as a trial-and-error approach by many parents, whereby they would gradually introduce different foods and monitor BGLs. 


*“The first meal we did was breakfast and, oh my god what a difference… we never did the normal cereals or anything like that. But we replaced it with an omelette. And oh my god, that just changed our day. Like, you know as soon as you start off with fabulous flat numbers, you can just carry on…”*
(Participant 6)

### 3.2. Promoters to Implementing a Low CHO Diet

Many parents cited the shock associated with diagnosis and difficulty maintaining glycaemic control as justification for the use of a low CHO diet following diagnosis.


*“…the first 6 months was just a complete blur of like not really knowing what on earth was going on.”*
(Participant 9)


*“as we just changed things slowly it just flattened right out. Yeah, so anyway that’s usually when I showed people when they say “why, why do you do that?” Like, this is why!”*
(Participant 5)

Parents stated that implementing a low CHO diet reduced their child’s need for insulin, making parents feel safer and more in control during the day and reducing the risk of hypoglycaemia at night.


*“They’re so much less at risk because you haven’t got this huge amount of insulin on board that’s going to, you know, cause some kind of disaster. You know, like a real emergency.”*
(Participant 6)

All parents reported that avoiding hyperglycaemia was a motivator for using a low CHO diet, explaining that hyperglycaemia was more stressful than hypoglycaemia and required more effort to manage.


*“Hypos are much easier to treat than hypers definitely. Phenomenally more easy and much quicker recovery as well.”*
(Participant 9)


*“…but nothing like the stress of trying to get a super high level down.”*
(Participant 5)

Parents focused on HbA1c as a measure of success in managing their child’s T1D. Parents reported worries associated with potential long-term complications associated with acute or sustained hyperglycaemia and identified feelings of guilt if their child experiences hyperglycaemia.


*“I think his last A1C was 6.3 before we went low carb and then that went to 5.3 as soon as we went low carb.”*
(Participant 6)


*“I feel guilty to be honest…. cause I hate to see him higher, and I worry about the damage that it’s doing to his body.”*
(Participant 15)

Parents commented that their own health improved since implementing a low CHO diet for their child. Benefits included reduced stress about BGLs, feeling empowered by the ability to keep BGLs stable and eating healthier food.


*“I feel so much better doing it, like I feel I’ve got so much more energy, and feel so much, like I lose weight and everything. For me if I eat more meat and vegetables without all the other stuff um, yeah, but it’s better for me, I know.”*
(Participant 4)

Parents reported their child’s emotions, behaviour and concentration was negatively impacted when BGLs were high, including children feeling sick, agitated and hyperactive. Most parents commented that their child’s behaviour was less erratic when using a low CHO diet and attributed this to having fewer glycaemic excursions that might negatively impact mood and behaviour.


*“…If he hits the number 6.0, or if he hits the number 3.8, he’s going to get emotional and upset and that’s not a good way to remember your birthday… let’s say you’ve got all your friends you’re in the middle of a party and you’re crying in the corner because you’re low or your high… that affects you emotionally.”*
(Participant 1)


*“She was such a grumpy bum. She’s so much happier and more relaxed and you know more confident when her levels are level.”*
(Participant 8)

Parents also described emphasising the link between their child’s BGLs and emotional state to try to encourage their child to make informed choices about food as they develop. 


*“…He will generally say “I think I’m a bit high, mum” and I say “do you think that’s why you’re a bit grumpy?” he goes “yeah”. And I say “you know what?” I said “that’s alright, we’ll just bring you down, what can we do?”*
(Participant 12)

### 3.3. Barriers to Implementing a Low CHO Diet

All parents acknowledged some challenges in implementing a low CHO diet but felt the benefits of stable BGLs, reduced stress and an increased feeling of control outweighed the challenges. The main barriers were the time required to plan and prepare meals, sourcing ingredients and food types that were low in CHO and the increased cost. 


*“…Another disadvantage is just that that level of organisation and, and sometimes the cooking, you know having to cook all the time, alternative type foods”.*
(Participant 7)


*“Probably the cost involved–um, ‘cause higher, our weekly shopping list has gone up quite a bit which might be hard for some families. Um, yeah for us we’re quite fortunate that yeah, we can be in that position.”*
(Participant 4)

Parents expressed a range of concerns regarding their child’s physical health, such as consuming too much fat or impaired growth, as well as the potential social and emotional consequences of a low CHO diet. Overall, parents were aware that low CHO diets were controversial, especially for children. 


*“I find that a lot of the recipes that I end up applying are for low keto diets, and I get a bit wary of keto… a lot of the time they are high in fat.”*
(Participant 11)


*“there’s guilt and a bit of doubt a bit of self-doubt that I’m maybe not doing the right thing–am I for this period of time stunting her growth? I don’t know.”*
(Participant 10)

Some parents expressed feelings of guilt related to limiting CHO for their children without T1D and reported offering high CHO options to them when their child with T1D was not present. 


*“We are very careful…my oldest is having his normal lunch box. And [child] never see me like, prepare. [Child] never sees me when I’m making it”*
(Participant 2)

Parents identified concerns associated with cultural norms of eating CHO containing foods. For instance, eating during holidays and at special occasions. Parents also reported worries about maintaining a low CHO diet during adolescence, when there is experimentation and greater emphasis on belonging. 


*“We’re sort of worried if he meets someone else with diabetes and they don’t eat the way he does.”*
(Participant 1)

Parents described putting effort into preparing foods that their children will enjoy. Many parents were aware that their children with T1D might feel they were missing out as a result of eating a low CHO diet and were mindful to include ‘treat’ foods in their diet to compensate for this. 


*“If we go to a birthday party or if I’m aware of some sort of treat happening at school, I will give her something … that she likes.”*
(Participant 10)

Parents noted that some challenges arose within the family when trying to meet the dietary needs of other family members. Despite these barriers, most parents perceived their child to enjoy the low CHO food offered. 


*“Everyone has learned to have new favourites–like we’ve got an almond porridge that all of the kids really love to eat…”*
(Participant 5)

### 3.4. Sources of Information on Low CHO Diets 

Parents sourced information from a range of sources including in-person and online, health professionals and peer support groups. Information from private diabetes services were reported to be more supportive of a low CHO approach, while clinicians at the paediatric diabetes services were perceived as not supporting low CHO diets for children with T1D. 


*“I have been too scared to even tell them we’re low-carb because of what happens–what other people say.”*
(Participant 3)


*“I’ve been told constantly by other parents just to not bring up low carb I’m saying this honestly, every single parents tell me to not bring up low carb with clinic, so the dietitians or anyone, just avoid it.”*
(Participant 8)

Parents reported that social media offered a safe place to discuss low CHO diets, as well as a course on implementing low CHO diets, and clinicians supporting their low CHO approach. One facility also offered in-person social interactions, including playgroups and education sessions.

Parents reported using online sources, including social media, video streaming channels and websites for information, recipes and for identifying low CHO alternatives. Additional sources of information included friends or family members with T1D.


*“I just shared how hard things were on Facebook one day and a friend of mine from college who has type one, she actually sent me a private message and asked if we’d ever heard about managing with low carb foods.”*
(Participant 5)


*“And it’s a group of low-carb, Type 1 Mums from all around the world and that is my saviour, that’s my group. We can, we laugh, we–fricken cry, we share appropriate jokes, and it really is my one.”*
(Participant 6)

Books were also sources of information, with Dr Bernstein’s book being the most reported, followed by diabetes-friendly recipe books.


*“I ordered that book [Diabetes Solution by Dr Bernstein] that same day because I just thought–we need some direction, we need to try something because what we’ve tried is not working.”*
(Participant 5)

## 4. Discussion

This study investigated the attitudes, knowledge and experiences of parents implementing a low CHO diet to manage their child’s T1D. Findings indicate parents were eager to speak about their experiences and open to sharing with researchers. One key finding was that parents’ understanding of a low CHO diet varied considerably, as did the foods and BGLs perceived as safe. Throughout the interviews, terminology around foods being ‘safe’ or a ‘threat’ were common. Parents typically referenced ‘safe’ foods as those that will not elevate BGLs, without reference to whether the food is beneficial for overall health, growth and development. 

There appeared to be a high level of anxiety associated with hyperglycaemia for most parents, with the restriction of CHO intake mitigating this risk. This highlights the need to address inaccurate risk perceptions related to daily BGL fluctuations. Parents described very narrow ‘acceptable’ BGL ranges and described experiences of distress when BGLs fell outside this range. The use of CGM may be contributing to this issue, as real-time data on BGL fluctuations following meals has shown to elicit anxiety and lead to subsequent food avoidance, regardless of the foods’ nutritional benefit [18,19]. In contrast, the longer-term nutritional risks associated with low CHO diets in children, such as excessive saturated fat intake, appear to be minimised or discounted [9]. Ensuring parents are supported to adopt appropriate and realistic daily glycaemic targets and manage distress associated with glycaemic excursions is vital.

Parents reported that their decision to adopt a low CHO approach was reinforced when HbA1c levels at clinic visits were below the recommended target of 53 mmol/mol (7.0%). However, parents reported not disclosing their child’s low CHO diet to clinicians as they believed it would not be supported, highlighting the need for greater engagement with families choosing low CHO diets. There is a need to better understand individual motivators and barriers underpinning dietary choices and the impact this may have on the diet quality of their child. 

Many parents reported the perceived link between high CHO foods, higher BGLs and their child’s emotional ability to be a promoter for adopting or maintaining a low CHO diet. Some parents reported explicitly highlighting this link to their child in an attempt to reinforce ‘healthy’ dietary choices. Supporting children to understand influences on their emotional state can aid in the development of emotion regulation skills [20]. However, parental reports in this study of linking emotions to food raises some concerns about the potential for children to misattribute fluctuations in mood to food and neglect other influences. This poses the risk that children may learn to regulate their emotions through food at the expense of other adaptive strategies. Adolescents with T1D are twice as likely to develop an eating disorder than their peers without diabetes [21] and restrictive eating practices are a known risk factor for the development of eating disorders in the general population. Among adults with T1D, breaking a perceived dietary ‘rule’, such as consuming more CHO than believed appropriate, is associated with an increased risk of insulin restriction, which is a hallmark of eating disorders in people with T1D [22,23]. It is unclear whether the use of low CHO diets in children with T1D is associated with an increased risk of developing an eating disorder; further research is needed to understand this relationship.

Parents were willing to exert high levels of effort in time, cost and organisation to maintain a low CHO diet. The salient promoters of more stable daily BGLs and reduced behavioural problems in their children reveal a tendency to favour short-term benefits over the potential longer-term risks to growth and development.

The health system is ‘numbers’ driven with a justifiable focus on HbA1c. Parents in this study highlighted that receiving feedback that their child’s HbA1c was within target range reinforced their decision to adopt a low CHO diet. However, reports from parents that they do not disclose their use of low CHO diets to their healthcare team raises concerns that the focus on ‘numbers’ may contribute to a failure to adequately address broader aspects of health (both physical and emotional) associated with T1D. Caregivers reported knowledge of health outcomes associated with hyperglycaemia and hypoglycaemia (resulting in an attitude of ‘the less insulin the better’ for some parents). However, knowledge around the health benefits of foods containing CHO, such as wholegrain cereals, grains and fruit, was either limited or concerns about growth and long-term health associated with restriction of CHO were minimised [24]. 

Lastly, greater psychosocial support for parents (particularly regarding insulin use) and a more inclusive approach from all health professionals in evidence-based clinical settings may reduce disengagement and/or increase transparency from families regarding dietary choices. In the future, a clinical protocol for low CHO diets and T1D may provide medical safety and monitoring (growth, CVD risk factors and eating behaviours) yet allow for the choice of parents to adopt a diet pattern that does not align with clinical guidelines [13]. A more individualised approach to support families’ preferences from diagnosis could reduce the likelihood of families seeking advice and support from unregulated and often non-evidence-based sources, such as social media groups. 

Limitations for this study include the small sample size and the potential for recruitment bias as the parents who agreed to participate in this study are already supportive of this dietary approach. Therefore, these results reflect a small, self-selected sample and further studies are recommended that explore the experience of families with older children and include an assessment of dietary intake, growth and knowledge about the potential adverse effects.

An additional limitation is that 36% of children reported to be on a low carbohydrate diet were on multiple daily injections compared to alternative pump or hybrid closed loop therapy, which afford tighter glucose control and may skew parental perceptions of impact of diet on blood glucose management compared to insulin technology [25]. With only thirteen families (*n* = 14) completing the interviews, despite efforts to recruit more, the research team, believed the dataset collected was rich as it was composed of specific experiences and knowledge, which was sufficient in addressing the narrow research aim with strong communication between researcher and participant [26].

## 5. Conclusions

These study findings have identified key parental concerns which appear to be influencing decisions around low CHO diets for children with T1D. Despite a lack of evidence-based guidelines supporting this dietary approach in children with T1D, all parents reported that the benefits outweigh the challenges. This study demonstrates the rich qualitative data that can be collected from families living with diabetes relating to nutrition—a key management strategy of T1D. Future areas of research to advance knowledge could include investigating how to best engage caregivers about recommended dietary practices; how to support accurate risk perception about BGL fluctuations; insulin use and dietary choices; and how to sensitively correct information that does not align with evidence-based practice. 

## Figures and Tables

**Figure 1 nutrients-15-01666-f001:**
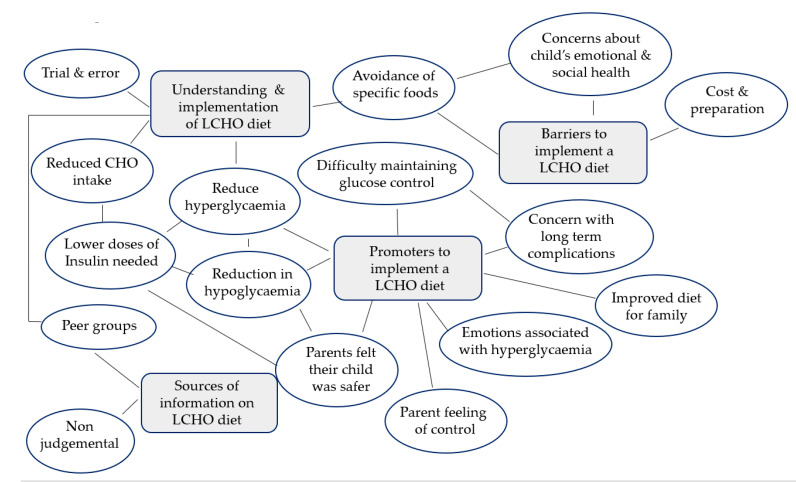
Thematic map of key findings.

**Table 1 nutrients-15-01666-t001:** Characteristics of parent participants (*n* = 14) and their children (*n* = 13).

Characteristic	n (%)	Mean ± St Dev (Range)
Participants (parents)	14 (100)	-
Mothers	13 (93)	-
Fathers	1 (7)	-
Age of participants (yr)	-	39.8 ± 6.5 (30.3–49.0)
**Education level**		
College	3 (21)	-
Undergraduate degree	9 (64)	-
Postgraduate degree	2 (14)	-
**Employment status**		
Full-time	2 (14)	-
Part-time	4 (28)	-
Casual	3 (21)	-
Unemployed	5 (36)	-
**Location**		
Metropolitan	12 (86)	-
Rural	2 (14)	-
Family history of type 1 diabetes (T1D)	2 (14)	-
**Child with T1D**		
Female	4	
Male	9	
Age (yr)	-	6.6 ± 2.0 (2–10)
Age at diagnosis (yr)	-	3.9 ± 1.7 (2.0–6.9)
Duration of T1D (yr)	-	2.7 ± 1.9 (0.6–6.5)
HbA1c %–participant self-reported	-	6.06 ± 0.91 (4.4–7.4)
HbA1c (mmol/mol)	-	42.75 ± 9.93 (24.6–57.4)
Continuous glucose monitor users	13 (93)	-
**Child Insulin Regimen**		
Pump	9 (64)	-
Multiple daily injections	5 (36)	-

## Data Availability

Data will be provided upon request.

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
