# Peer review of "Experiences and Attitudes of Parents Reducing Carbohydrate Intake in the Management of Their Child’s Type 1 Diabetes: A Qualitative Study"

_nutrients, 2023, doi:10.3390/nu15071666_

Round 1

Reviewer 1 Report

Summary:  This is a very well written manuscript of a qualitative interview study of caregivers who have implemented low carb diets and share their information sources, challenges and successes. The results are limited by the limited perspective in that it seems that most interviewees supported a low-carb diet and few had anything negative to say portraying only the positive aspect of LCDs, but this may also be particular to the population of respondents. Given the increasing use of low-carb diets, this is an important contribution to the existing literature in understanding caregivers perspective, especially for providers who may rely on literature that only shows the negative aspects of low-carb diets in T1D.  The manuscript would be strengthened by an additional table or figure that summarizes some of the highlights of the report.

Introduction:   Well written background

Methods:

Very clear and concise methods.

Results:

Participant demographics – was household income or a surrogate measure of that information available? 

It is interesting that 1/3 of the participants are on MDI. Were any of the children of the participants on a HCLS?  The improved glycemic control afforded by these systems may impact how necessary parents feel about a low-carb diet and this should be mentioned as a limitation in the discussion.

Did parents elaborate on why they did not share with their diabetes team that they were implementing a low-carb diet? Prior negative experiences or interactions related to this topic?  Counseling on avoidance of a low-carb diet?

Discussion: Conclusion paragraph nicely states the overall summary of the manuscript

Table 1 – include household income or insurance information.

 The manuscript would be strengthened by an additional table or figure that summarizes some of the highlights of the report.  

Author Response

Reviewer One

We wish to thank Reviewer One for their insightful comments and suggestions. We agree with Reviewer One that this is a popular and emerging area of the nutrition management of paediatric diabetes and caregivers’ perspectives are important, especially when looking at the dietary intake of young children.

Please see our responses to Reviewer One’s comments below.

Participant demographics – was household income or a surrogate measure of that information available? 

The research team agree with Reviewer One that household income would have been useful information to collect from participants. This data was not collected from participants at time of data collection, however, surrogate measures of socio-economic status, including education level were collected. As seen in Table 1, 78% of participants had an undergraduate or postgraduate university education. More than one third of the participants reported being unemployed. This does not reflect the general Australian population, although may highlight the required time commitments and unpaid work caring for a child with type 1 diabetes. The authors agree than household income is important to collect in future studies.

No edits have been made to the manuscript as a result of this comment.

It is interesting that 1/3 of the participants are on MDI. Were any of the children of the participants on a HCLS?  The improved glycemic control afforded by these systems may impact how necessary parents feel about a low-carb diet and this should be mentioned as a limitation in the discussion.

Thank you for this comment. The participants were selected for the study based on caregiver reports of implementing a low carbohydrate diet, and insulin regime was not an inclusion criteria. However, only one of the thirteen children of participants was using a hybrid closed loop system at the time of the qualitative interview.

The following limitation has been added to the discussion (lines 373-380) to reflect this comment.

Additionally, 36% of participants were on multiple daily injections, compared to alternative pump or hybrid closed loop therapy, which affords tighter glucose control and therefore may skew parental perceptions of impact of diet on blood glucose management compared to insulin technology (25).’

Did parents elaborate on why they did not share with their diabetes team that they were implementing a low-carb diet? Prior negative experiences or interactions related to this topic?  Counselling on avoidance of a low-carb diet?

Reported experiences varied and ranged from having direct negative interactions with clinicians who were openly against a low carbohydrate diet, to other parents via online or community groups suggesting they not mention low carbohydrate diets in clinic. All children in Western Australia are under the medical management of one tertiary paediatric hospital.

To elaborate on this point and enhance understanding by the reader, we have included two additional quotes in the amended manuscripts on lines 280-284.

“I have been too scared to even tell them we’re low-carb because of what happens – what other people say.” (Participant 3)

“I've been told constantly by other parents just to not bring up low carb I'm saying this honestly, every single parents tell me to not bring up low carb with clinic, so the dietitians or anyone, just avoid it.” (Participant 8)

The results are limited by the limited perspective in that it seems that most interviewees supported a low-carb diet and few had anything negative to say portraying only the positive aspect of LCDs, but this may also be particular to the population of respondents

The inclusion criteria specifically included caregivers who were currently or had recently implemented a low carbohydrate diet to manage their child’s diabetes. We acknowledge that the findings would have been strengthened if participants who had tried and discontinued with this dietary approach volunteered; however, participation was voluntary, and we exhausted recruitment avenues. The participants who volunteered for the study were open and willing to share their clinical experiences anonymously for the first time.

Additional text has been included in the limitations section of the discussion to remind the reader of the self-selected sample on line 371.

“Limitations for this study include the small sample size and the potential for recruitment bias as the parents who agreed to participate in this study are already supportive of this dietary approach. Therefore, these results reflect a small, self-selected sample.”

Table 1 – include household income or insurance information.

Household income and insurance data were not collected during this study. All participants recruited are patients of the only paediatric tertiary hospital in Western Australia, who diagnoses all children in the state diagnosed with type 1 diabetes. The Australian public health care system is free and provides ongoing diabetes care at no cost to families.

No edits have been made to the manuscript as a result of this comment.

The manuscript would be strengthened by an additional table or figure that summarizes some of the highlights of the report.  

We agree that our manuscript would be strengthened by a figure that summarises the highlights of the report, and therefore we have added a figure of a thematic map of the key themes that emerged from the qualitative interviews. Thank you for this suggestion.

Figure 1 has been added to the manuscript and now appears on line 138.

Reviewer 2 Report

I would like to thank you for the opportunity to review this manuscript entitled "Experiences and attitudes of parents reducing carbohydrate intake in the management of their child’s type 1 diabetes: A qualitative study".
The topic is of interest I modestly think that is always good to remember that in this research area it is important to not forget how parents and young people with T1DM interpret the clinical suggestions and how this impact their lives. Therefore the work is of interest, however since of the very small sample size, I think it could be important to add in the title "a pilot qualitative study", or in any case, the conclusions should be taken more cautiously considering the absolutely preliminary nature of these findings.

In addition, I think that when discussing CHOs strategies in T1DM, especially in the youth, it should be important to ask how parents manage CHOs intake for example in preparation for physical exercise or sport, which should be encouraged already during the first years from the diagnosis since often T1DM diagnosis is seen as a barrier (10.2337/dc08-0720). It should be also important to discuss this point further, as also smart technology and smartphone applications can support parents and people with T1DM to optimize CHOs intake to promote safe and optimal physical activity (10.1016/j.jcjd.2016.09.007; 10.1089/dia.2011.0052;10.3390/nu11123017).

Author Response

Reviewer Two

We would like to thank Reviewer Two for their knowledge and suggestions. The research team have had engaging conversations about potential areas for future research because of these helpful comments, specifically related to physical activity and the adoption of diabetes technology. We agree that as a scientific community we must be reminded and exposed to the lived experiences of patents and young people living with type 1 diabetes, and how clinical recommendations are interpreted and implemented in the community.

I think it could be important to add in the title "a pilot qualitative study", or in any case, the conclusions should be taken more cautiously considering the absolutely preliminary nature of these findings.

Thank you for your suggestion to cautiously consider the findings due to the small sample size. We agree that the conclusions need cautious consideration and in response, have included the following text in the limitations section of the discussion (Line 373).

‘Therefore, these results reflect a small, self-selected sample and recommend further studies that explore the experience of families with older children and include an assessment of dietary intake, growth, and knowledge about the potential adverse effects.’

The authors would like to acknowledge that although the sample size is small and this needs to be considered, the sample size of n=14 was adequate for a qualitative study to answer the research question and gain insight into why parents choose to implement a low carbohydrate approach to help manage their child’s blood glucose levels. The research team exhausted recruitment avenues within the Western Australian Diabetes Database, that includes the caregivers of all children diagnosed with type 1 diabetes in Western Australia (approximately n=1200). No follow up qualitative study is planned and therefore, this study does not constitute a pilot.  For this reason, the authors agreed that including additional text (as above) would suffice and ‘pilot’ has not been added to the title of the manuscript.

In addition, I think that when discussing CHOs strategies in T1DM, especially in the youth, it should be important to ask how parents manage CHOs intake for example in preparation for physical exercise or sport, which should be encouraged already during the first years from the diagnosis since often T1DM diagnosis is seen as a barrier (10.2337/dc08-0720). It should be also important to discuss this point further, as also smart technology and smartphone applications can support parents and people with T1DM to optimize CHOs intake to promote safe and optimal physical activity.

Thank you for the suggestion and references. We agree that dietary choices cannot be considered in isolation, and they’re closely related to and motivated by physical activity, mental health, glycaemic outcomes and more.

Exercise and diabetes technology were not directly discussed in these interviews as the research question was specially related to the knowledge, attitudes and experiences of using low carbohydrate diets. However, the authors agree these are important topics to include in future qualitative interviews we conduct in type 1 diabetes, especially because both exercise and technology can be related to the use of low carbohydrate diets.

As a result of this suggestion, we reviewed the interview transcriptions and interestingly, managing exercise while restricting carbohydrate intake was not identified as barrier when asked to report on any challenges or barriers to a low carbohydrate diet.

Regarding the use of technology to support caregivers to optimise carbohydrate intake, we agree and this is recognised by the clinical team we work closely with at Perth Children’s Hospital. The qualitative interviews in the present study were designed not to provide participants with solutions but rather listen and document their knowledge, attitudes and lived experiences. 

No changes to the manuscript have been made as a result of this comment.

Round 2

Reviewer 2 Report

I am fine with the authors' responses. Thank you.